# Glycemic profile and associated factors in indigenous Munduruku, Amazonas

**Hanna Lorena Moraes Gomes** [1] *, **Neuliane Melo Sombra**[1], **Eliza Dayanne de Oliveira Cordeiro**[1], **Zilmar Augusto de Souza Filho**[1], **Noeli das Neves Toledo**[1], **Evelyne Marie Therese Mainbourg** [2], **António Manuel Sousa**[3], **Gilsirene Scantelbury de Almeida**[1]

**1** Manaus School of Nursing, Federal University of Amazonas, Manaus, Brazil, **2** Leônidas & Maria Deane Institute / FIOCRUZ Amazônia, Manaus, Brazil, **3** Amazonas State University, Manaus, Brazil

* hannahlorena.mg@gmail.com

**Data Availability Statement:** All relevant data are within the manuscript.

**Funding:** This study received funding approved by the National Council for Scientific and

## Abstract

### Objective

To evaluate the glycemic profile and its association with sociodemographic, anthropometric, clinical and lifestyle factors of Munduruku indigenous people.

### Method

Cross-sectional study with a quantitative and analytical approach, a total of 459 indigenous people (57.1% men, aged 36.3 ± 14.7 years old) belonging to the Munduruku ethnic group from the Kwatá-Laranjal Indigenous Land, in Amazonas, Brazil, were selected by probabilistic sampling in all households in the four most populous villages. Sociodemographic and anthropometric variables, blood pressure levels and lipid profile were evaluated. Fasting capillary blood glucose was measured with a digital device. The associations were assessed by multinomial logistic regression, and p-values≤0.05 were considered significant.

### Results

For pre-diabetes, prevalence was 74.3% and, for diabetes, 12.2%. The variables associated with the risk for pre-diabetes were the following: age (OR = 1.03; 95% CI = 1.00 – 1.06) and obesity (OR = 9.69; 95% CI = 1.28 – 73.58). The positive associations indicating risk for diabetes were as follows: age (OR = 1.05; 95% CI = 1.03 – 1.08), overweight (OR = 4.17; 95% CI = 1.69 – 10.32) and obesity (OR = 35.26; 95% CI = 4.12 – 302.08).

### Conclusions

The risks associated with pre-diabetes and diabetes among the Munduruku indigenous people revealed a worrying index. It is necessary to consider changes in eating habits and lifestyle, as well as possible environmental and social changes that can affect this and other groups, with emphasis on those who live in vulnerable conditions.

Technological Development (CNPq) (Proc. 424053 / 2016-0) and with funding from the Scientific Article Publication Support Program (PAPAC) and the Post Support Program -Graduation (PROSGRAD), both of these are programs of the Amazonas Research Support Foundation - FAPEAM. Funders had no role in the study design, data collection, analysis, decision to publish or preparation of the manuscript.

**Competing interests:** The authors have declared that no competing interests exist.

## Introduction

The changes in the globalized world, as a result of the urbanization and industrialization processes, brought about changes in habits and lifestyles, contributing to the increase of chronic non-communicable diseases, among which we can highlight cardiovascular diseases. These same impacts permeate the indigenous populations, through transitions in life, economic and sociocultural habits, and in their own lifestyle [1, 2].

The destruction of the ecosystems that the Brazilian Indigenous Lands are facing, together with the acceleration of the urbanization process, sedentary lifestyle, changes in the diet, obesity and easy access to cities, contribute significantly to the transformations of the daily lives of indigenous populations, leaving them more vulnerable to certain diseases, which contributes to the increase of Chronic Non-communicable Diseases (CNCDs) [3, 4].

Social indicators of a national scope classify the North Region as belonging to Class "E" of social vulnerability, as it consists of extensive rural areas, low demographic density, with a very low human development index, precarious access to treated water, sewage and electricity, among other negative results. Compared with the South and Southeast regions of the country, the North has less capacity to respond to health problems, in terms of Health Care Network structure [5].

Deaths due to non-communicable diseases (NCDs) represented the highest percentage: 73.4% (95% uncertainty interval [UI] = 72.5 – 74.1) in 2017. In relation to 2007, there was a 22.7% (21.5 – 23.9) increase, equivalent to 7.21 million (7.20 – 8.01) of estimated additional deaths. There was a major increase in years of life lost due to neoplasms and cardiovascular diseases.

In the general population, cardiovascular diseases (CVDs) are part of the group of main causes of mortality. In 2016, approximately 17.9 million people died due to CVDs worldwide. From this perspective, diabetes mellitus (DM) stands out as a highly prevalent health problem and one of the main risk factors for CVDs [6–8].

DM is configured as a "metabolic disorder" characterized by persistent hyperglycemia, resulting from a deficit in the production of insulin or in its action, or even in both mechanisms, leading to long-term complications" (SBD, pg. 19). Data from the International Diabetes Federation point out that, in the world, 8% of adults lived with DM in 2017. DM is a growing and important health problem that affects the population of all countries, being responsible for 4 million deaths worldwide in a single year [9, 10].

It is believed that changes in the social, economic and political scopes of indigenous Brazilians may have favored changes in their lifestyle and in their epidemiological profile [1]. In the Brazilian indigenous population, the first cases of DM began to be investigated from the 1970s, when the prevalence of diabetes was non-existent [1].

In the state of Mato Grosso do Sul, several studies were carried out with the Terena, Guarani and Kaiowá indigenous peoples, where it was found that 4.5% had DM in 2007 and 2008 [11]. Another two studies carried out in the same population found a prevalence rate of 5.8% in the period from 2009 to 2011, and of 4.5% in 2008 and 2009 [12]. In 2013, among 385 Terena and Guarani women from the same region, 7% presented altered capillary glycaemia suggestive of DM [13]. Among the Guarani and Tupinikin (ES), in 2003 and 2004, the prevalence of DM was 4.5% [14]. In Khisêdjê in 2010 and 2011, prevalence was 3.8% [15]. The highest prevalence rate of DM among indigenous people in Brazil was found among the Xavante in the state of Mato Grosso (n = 948): 25.9% [16].

The data presented show that diabetes has been growing in indigenous populations [17] and that is worsened by the increased consumption of industrialized food products, social problems linked to the economy and the increasingly frequent contact with the non-

indigenous population [1, 17]. Considering that most of the studies refer to ethnicities in the Brazilian Midwest Region, the objective of the study was to assess the glycemic profile and its association with sociodemographic, anthropometric, clinical and lifestyle factors of Munduruku indigenous people from the state of Amazonas, Brazilian North Region.

## Method

### Study locus and population

The study was carried out in the Kwatá-Laranjal indigenous land (Fig 1), located in the municipality of Borba, state of Amazonas, in the Brazilian North Region. The study population consisted of individuals from the Munduruku ethnic group who live in the villages of Laranjal, Mucajá, Kwatá and Fronteira, members of the Kwatá-Laranjal Indigenous Land, aged between 18 and 80 years old, and of both genders. According to population data, released by the Special Indigenous Sanitary District of Manaus in 2018, the total population over 18 years old of both genders living in these four villages consisted in 635 inhabitants, divided as follows: 195 in Mucajá, 118 in Laranjal, 186 in Kwatá and 136 in Fronteira.

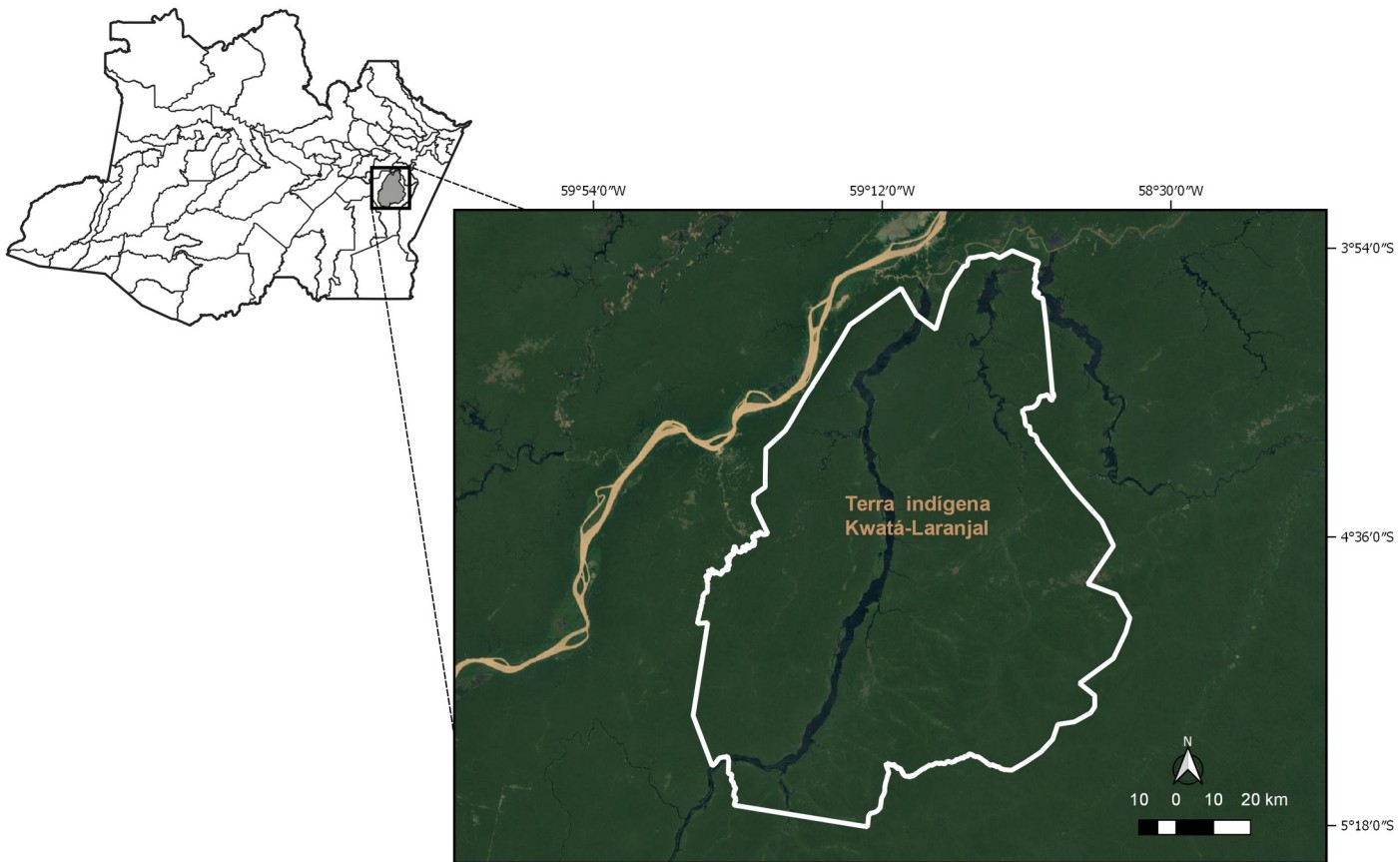

DATUM: SIRGAS 2000 (EPSG: 4674)          Data de atualização: 16/03/2021

Elaborado por: NAP/ILMD - Fiocruz Amazônia

**Fig 1. Geographic location of the Kwatá-Laranjal Indigenous Land.**

## Study participants

The following was accepted for sample calculation: 50.0% proportion of the indigenous population and the prevalence values of diabetes pointed out by the Guidelines of the Brazilian Diabetes Society and by the study by Soares et al. [9, 18]. The error margin adopted was 5%, 95% confidence interval, and 10% for losses. The sample consisted of 459 individuals belonging to the Munduruku ethnicity, from the villages of Mucajá (n = 129), Laranjal (n = 93), Kwatá (n = 136) and Fronteira (n = 101).

The four most populous villages in the Kwatá-Laranjal Indigenous Land (Mucajá, Laranjal, Kwatá and Fronteira) were chosen. Probabilistic sampling of individuals per household was carried out, in which all members had an equal chance of participating in the study. The study included indigenous people belonging to the Munduruku ethnic group, aged ≥ 18 years old and living in the selected villages. It is noted that all the Munduruku indigenous people drawn to participate in this study were able to fluently communicate in the Portuguese language. Only those who were ill and pregnant were excluded from the sample.

## Data collection

Before starting data collection in the Kwatá-Laranjal Indigenous Land, the team of women researchers visited the four villages included in this study, which allowed for previous contact with the local indigenous leaders, closer contact with the health professionals who served in those villages, and holding a meeting with the indigenous people to present the research objectives and method.

For the data collection stage, the team underwent specific training in order to standardize the procedures for: measuring blood glucose and capillary lipids after fasting for a minimum of eight hours, measuring blood pressure, taking anthropometric measurements and conducting the interview.

At the beginning of data collection, the residents were invited again to be informed about how the participants would be selected and the procedures for data collection. For each household, the research participants were selected by means of a draw. The indigenous health agent assisted the team in locating the homes of the selected participants. The guidelines for data collection were given the day before, with reinforcement regarding the location, day, time and, mainly, the need for at least 8-hour fasting.

The collection of the anthropometric data, blood pressure, blood glucose and lipids was always performed at dawn. Before starting the collection of blood drops from the digit pulp, the indigenous people were asked at what time they had their last meal. Those who reported breaking the fast were rescheduled for the following day and re-oriented.

In relation to the tests of capillary blood glucose and lipid levels, the equipment used were as follows: *Active* portable digital device from the Accu-Chek® manufacturer for the measurement of capillary blood glucose and the Accutrend® *Plus* device for the measurement of cholesterol and triglycerides, both manufactured by Roche Diagnóstica, with their respective test strips (Accutrend® Cholesterol and Accutrend® Triglycerides). The cut-off points used to assess and classify fasting capillary glucose were as follows: normal < 100 mg/dL, pre-diabetes ≥ 100 mg/dL and < 126 mg/dL and diabetes ≥ 126 mg/dL [9]. For the lipid levels, the classification was the following: hypercholesterolemia when ≥ 240 mg/dL and hypertriglyceridemia when ≥ 175 mg/dL [19].

The following was used for the evaluation of the anthropometric measures: digital bioimpedance scale (OMRON HBF-514C), portable stadiometer (ALTURA EXATA) and inelastic measuring tape. The neck circumference measurement was taken at the smallest neck circumference, just above the laryngeal prominence. The waist circumference measurement was

taken at the midpoint between the last rib and the lateral iliac crest, around the narrowest part of the trunk. The taper index was determined, according to its definition, from the measurements of weight, height and waist circumference. Both BMI and Body Fat Percentage were assessed using the bioimpedance technique.

The cut-off points adopted to classify neck circumference measurements were as follows: $\geq$ 37 cm for men and $\geq$ 34 cm for women; and those for waist circumference were: $\geq$ 102 cm for men and $\geq$ 88 cm for women [20]. For the taper index, the adopted values were: $\geq$ 1.25 for men and $\geq$ 1.18 for women. As for the Body Mass Index (BMI), it was classified as: low weight ($<$ 18.5 kg/m$^2$), normal weight (18.5 kg/m$^2$-24.9 kg/m$^2$), overweight (25.0 kg/m$^2$-29.9 kg/m$^2$) and obesity ($\geq$ 30.0 kg/m$^2$) [21]. The classification of body fat percentage considered the following stratification by age group and gender: low ($<$ 8.0%-$<$ 13.0% for men and $<$ 21.0%-$<$ 30.0% for women), normal (13.0%-24.9% for men and 30.0%-$\leq$ 35.9% for women) and high ($\geq$ 25.0% for men and $\geq$ 36.0% for women).

Blood pressure levels were measured on the left arm, using an automatic professional blood pressure monitor (OMRON/Model HBP-1100), properly calibrated. The procedures to perform the measurement and classification of blood pressure were conducted according to the Brazilian Hypertension Directive. The following cut-off points were considered: pre-hypertension when systolic blood pressure levels are between 140 mmHg and 159 mmHg and/or when the diastolic blood pressure is between 90 mmHg and 99 mmHg; hypertension when the value is $\geq$ 180 mmHg in systolic pressure and/or $\geq$ 110 mmHg in diastolic pressure. Alternatively, hypertension could be self-reported, if the indigenous participants reported having been diagnosed with hypertension by a physician or if they were taking some antihypertensive medication, regardless of the blood pressure values measured in the interview [22].

For the assessment of lifestyle, the level of physical activity was investigated using the IPAC (International Physical Activity Questionnaire), in its short version, an instrument validated with translation into the Portuguese language. The IPAQ allows quantifying the total minutes spent in weekly physical activities and surveying the distribution of time by intensity of the physical activity practiced. The level of physical activity was classified according to the instrument. To assess the intake of alcoholic beverages, the Alcohol Use Disorder Identification Test (AUDIT) questionnaire was used, allowing the identification of risk and harmful consumption and of probable dependence on alcohol in the past 12 months.

A form consisting of closed questions related to the following variables was applied: gender, age, marital status, schooling, paid work, social benefit received, monthly family income, self-reported hypertension and/or consumption of antihypertensive medications, smoking, level of physical activity, alcohol consumption and family history of cardiovascular diseases.

The participants who presented changes in capillary glycaemia, triglycerides, total cholesterol or/and blood pressure, as well as those who were obese were referred directly to the care provided by the health team at the Base Center (reference health unit, belonging to the Indigenous Health Sub-System) for evaluation and monitoring. For the changes in the anthropometric markers, this information was passed on to the health professionals working in the respective Base Center.

## Statistical analysis

The analysis of the data collected was performed by means of the R software, version 3.5.1. The Kolmogorov-Smirnov test was used to verify normal distribution of the data. In this way, the continuous variables were presented using means and standard deviations; and the categorical variables, with absolute and relative frequencies. For the continuous variables, the Kruskal-Wallis test was used; and for categorical ones, Fisher's Exact test. The significance level was set

at 5%. The Wald test was used for the multinomial logistic regression analysis. To verify the association between the dependent variables (pre-diabetes and diabetes) and the independent variables of the study, Odds Ratios (ORs) were estimated based on the multinomial regression model and the respective 95% confidence interval (CI). For this being a multifactorial phenomenon, the independent variables were grouped in blocks (sociodemographic, lifestyle and anthropometric and clinical factors) and analyzed hierarchically.

## Ethical aspects

The data were collected from August to September 2018, after the consent of the leaders of the Kwatá-Laranjal Indigenous Land, approval by the National Research Ethics Commission (CAAE 74361617.2.0000.5020), and authorization for entry into indigenous lands of the Ministry of Justice National Indian Foundation (43/AAEP/PRES/2018). All the indigenous people who agreed to participate in the study signed the Free and Informed Consent Form.

## Results

As shown in Table 1, the profile of the glycemic levels of the 459 indigenous Munduruku indicates that 86.5% had high serum levels of fasting capillary glycaemia, with 74.3% being suggestive of pre-diabetes and 12.2% of diabetes.

As for the sociodemographic factors, it was observed that 57.1% were men, with a mean age of 36.6 years old, most with a partner, and 9.6% not having any schooling level. A little over half of them were unemployed and 61.7% received some social benefit from the Brazilian federal government. In this way, most of the Munduruku indigenous people had a monthly family income of up to US$ 470.67.

The general anthropometry assessment allowed identifying that the indigenous people had high mean values of neck circumference, waist circumference and taper index. The mean BMI indicated excess weight, in addition to the majority presenting high body fat percentages.

In relation to the clinical factors of the Munduruku indigenous people, the mean pressure levels indicated normality, but 10.2% presented high levels of systolic and diastolic blood pressure, suggestive of hypertension. Regarding the serum triglyceride levels, the indigenous population presented a high mean value but, for total cholesterol, the mean remained within normal limits.

Regarding the indigenous people's lifestyle, there was a high prevalence of alcohol consumption (71.2%) and smoking (54.2%), as well as a low prevalence of sedentary lifestyle (7.6%). It is worth mentioning that most of the indigenous people reported having a family history of hypertension and diabetes.

Table 1 also shows that the group of Munduruku indigenous people with diabetes presented statistically significant differences when compared to the other groups, in greater proportion having some paid work and, in a smaller proportion, receiving some social benefit. The group of diabetics presents higher values regarding age, mean in the anthropometric markers, prevalence of obesity and body fat, prevalence of pre-hypertension and hypertension, mean of triglycerides and total cholesterol, as well as family members with diabetes or/and stroke.

Table 2 shows the unadjusted multinomial logistic regression model. The association of pre-diabetes with age showed that, for every one-year-old increase in the age of the indigenous Munduruku, their chance of becoming pre-diabetic increases by 4%. It is also worth noting that the indigenous people without a partner had a lower risk of being pre-diabetic (OR = 0.55 [95% CI = 0.32 – 0.96]).

As for the association of pre-diabetes with the anthropometric factors, it was observed that, with a one-centimeter increase in waist circumference (OR = 1.07 [95% CI = 1.03 – 1.10]), in

**Table 1. Categorization of the glycemic profile of indigenous Munduruku according to the sociodemographic and anthropometric variables, clinical factors, habits and lifestyle, and family history.**

| Variables | Glycemic Profile | | | | p-value |
|---|---|---|---|---|---|
| | Normal | Pre-diabetes | Diabetes | Total | |
| | N (%) | N (%) | N (%) | N (%) | |
| | 62 (13.5) | 337 (74.3) | 60 (12.2) | 459 (100) | |
| *Sociodemographic Factors* | | | | | |
| **Gender** | | | | | 0.404 |
| Female | 22 (35.5) | 147 (43.6) | 28 (46.7) | 197 (42.9) | |
| Male | 40 (64.5) | 190 (56.4) | 32 (53.3) | 262 (57.1) | |
| Age (years old), mean (SD) | 30.2 (±11.2) | 36.5 (±14.8) | 44.1 (±14.0) | 36.6 (±14.7) | **<0.001** |
| **Marital Status** | | | | | 0.109 |
| Has a partner | 35 (56.5) | 236 (70.0) | 41 (68.3) | 312 (68.0) | |
| No partner | 27 (43.5) | 101 (30.0) | 19 (31.7) | 147 (32.0) | |
| **Schooling** | | | | | 0.116 |
| Illiterate | 3 (4.8) | 30 (8.9) | 11 (18.3) | 44 (9.6) | |
| Elementary School | 19 (30.6) | 134 (39.8) | 22 (36.7) | 175 (38.1) | |
| High School | 30 (48.4) | 128 (38.0) | 19 (31.7) | 177 (38.6) | |
| Higher Education or Postgraduate | 10 (16.1) | 45 (13.4) | 8 (13.3) | 63 (13.7) | |
| **Paid work** | | | | | **0.039** |
| Yes | 22 (35.5) | 138 (40.9) | 34 (56.7) | 194 (42.3) | |
| No | 40 (64.5) | 199 (59.1) | 26 (43.3) | 265 (57.7) | |
| **Social benefit** | | | | | **0.045** |
| Yes | 43 (69.4) | 211 (62.6) | 29 (48.3) | 283 (61.7) | |
| No | 19 (30.6) | 126 (37.4) | 31 (51.7) | 176 (38.3) | |
| **Monthly family income (minimum wage[a])** | | | | | 0.708 |
| Does not have | 21 (26.2) | 7 (2.2) | 1 (1.7) | 29 (6.3) | |
| <1 minimum wage (US$ 235.34) | 30 (37.5) | 134 (41.7) | 21 (36.2) | 185 (40.3) | |
| 1 - 2 minimum wages (US$ 235.35 – US$ 470.67) | 22 (27.5) | 115 (35.8) | 24 (41.4) | 161 (35.1) | |
| 3 - 4 minimum wages (US$ 706.01 – US$ 941.35) | 6 (7.5) | 49 (15.3) | 11 (19.0) | 66 (14.4) | |
| ≥ 5 minimum wages (US$ 1,176) | 1 (1.3) | 16 (5.0) | 1 (1.7) | 18 (3.9) | |
| *Anthropometric Factors* | | | | | |
| **Neck circumference (cm), mean (SD)** | 35.5 (±3.3) | 36.2 (±3.3) | 37.7 (±3.2) | 36.3 (±3.3) | **0.001** |
| **Waist circumference (cm), mean (SD)** | 79.5 (±7.9) | 85.1 (±10.1) | 92.2 (±8.7) | 85.3 (±10.2) | **<0.001** |
| **Taper index, mean (SD)** | 1.20 (±0.08) | 1.24 (±0.09) | 1.29 (±0.07) | 1.24 (±0.09) | **<0.001** |
| **BMI (kg/m$^2$), mean (SD)** | 23.6 (±2.8) | 25.7 (±4.0) | 28.0 (±3.6) | 25.8 (±4.0) | **<0.001** |
| **BMI classification** | | | | | **<0.001** |
| Low weight (< 18.5 kg/m$^2$) | 1 (1.6) | 4 (1.2) | 0 (0) | 5 (1.1) | **< 0.001** |
| Normal weight (18.5–24.9 kg/m$^2$) | 42 (67.7) | 158 (46.9) | 12 (20.0) | 212 (46.2) | |
| Overweight (25.0–29.9 kg/m$^2$) | 18 (29.0) | 127 (37.7) | 31 (51.7) | 176 (38.3) | |
| Obesity (≤30 kg/m$^2$) | 1 (1.6) | 48 (14.2) | 17 (28.3) | 66 (14.4) | |
| **Body fat classification** | | | | | **<0.001** |
| Low (<8.0%-<13.0% men/<21.0%-<30.0% women) | 1 (1.6) | 7/337 (2.1) | 0 (0) | 8 (1.7) | |
| Normal (13.0%-24.9% men/30.0%-≤35.9% women) | 35 (56.5) | 126 (37.4) | 7 (11.7) | 168 (36.6) | |
| High (≥25.0% men; ≥36.0% women) | 26 (41.9) | 204 (60.5) | 53 (88.3) | 283 (61.7) | |
| *Clinical Factors* | | | | | |
| **Systolic blood pressure, SBP (mmHg), mean (SD)** | 110.0 (±12.2) | 113.6 (±15.0) | 121.2 (±16.9) | 114.1 (±15.2) | **<0.001** |
| **Diastolic blood pressure, DBP (mmHg), mean (SD)** | 63.6 (±8.2) | 66.5 (±8.4) | 70.4 (±8.8) | 66.6 (±8.6) | **<0.001** |
| **Blood pressure classification** | | | | | **0.001** |

*(Continued)*

**Table 1.** (Continued)

| Variables | Glycemic Profile | | | | p-value |
|---|---|---|---|---|---|
| | Normal | Pre-diabetes | Diabetes | Total | |
| | N (%) | N (%) | N (%) | N (%) | |
| | **62 (13.5)** | **337 (74.3)** | **60 (12.2)** | **459 (100)** | |
| Normal (SBP of ≤120–129 mmHg/DBP ≤80–84 mmHg) | 59 (95.2) | 294 (87.2) | 40 (66.7) | 393 (85.6) | |
| Pre-hypertension (SBP of ≥130 mmHg-139 mmHg/BPD ≥80–89 mmHg) | 1 (1.6) | 12 (3.6) | 6 (10.0) | 19 (4.1) | |
| Hypertension (SBP of ≥140 mmHg/DBP ≥90 mmHg) | 2 (3.2) | 31 (9.2) | 14 (23.3) | 47 (10.2) | |
| **Triglycerides (mg/dL)** | 131.9 (±65.8) | 149.3 (±86.3) | 206.8 (±124.1) | 154.5 (±92.1) | **<0.001** |
| **Total cholesterol (mg/dL)** | 171.3 (±25.5) | 176.5 (±32.3) | 189.7 (±35.4) | 177.5 (±32.2) | **0.003** |
| *Lifestyle* | | | | | |
| **Smoker** | | | | | 0.542 |
| Yes | 36 (58.1) | 184 (54.6) | 29 (48.3) | 249 (54.2) | |
| No | 26 (41.9) | 153 (45.4) | 31 (51.7) | 210 (45.8) | |
| **Level of physical activity** | | | | | 0.125 |
| Sedentary | 2 (3.2) | 26 (7.7) | 7 (11.7) | 35 (7.6) | |
| Irregularly active | 13 (21.0) | 92 (27.3) | 18 (30.0) | 123 (26.8) | |
| Active | 21 (33.9) | 114 (33.8) | 24 (40.0) | 159 (34.6) | |
| Very active | 26 (41.9) | 105 (31.2) | 11 (18.3) | 142 (30.9) | |
| **Alcohol Consumption** | | | | | 0.800 |
| Low risk consumption | 7 (25.9) | 35 (30.2) | 4 (23.5) | 46 (28.8) | |
| Risk intake, harmful or probable dependence | 20 (74.1) | 81 (69.8) | 13 (76.5) | 114 (71.2) | |
| *Family history* | | | | | |
| Hypertension | 43 (84.3) | 211 (77.0) | 43 (86.0) | 297 (79.2) | 0.222 |
| Diabetes | 31 (63.3) | 161 (61.7) | 38 (82.6) | 230 (64.6) | **0.023** |
| Stroke | 20 (45.5) | 75 (31.6) | 19 (52.8) | 114 (36.0) | **0.018** |

Kwatá-Laranjal Indigenous Land, Borba, Amazonas, Brazil, 2018.

[a] Current minimum wage of R$ 954.00, equivalent to approximately US$ 235.34 in August 2018.

the taper index (OR = 1.06 [95% CI = 1.02 – 1.09]) and in the BMI (OR = 1.20 [95% CI = 1.10 – 1.32]), the indigenous people have a chances to develop pre-diabetes of 7%, 6% and 20%, respectively. Excess weight among the indigenous people also presented an association with pre-diabetes, since the chance of the indigenous person who presented overweight to become pre-diabetic is 87%; and, among those who were obese, the chance becomes 12 times greater (OR = 12.76 [95% CI = 1.71 – 95.26]). For the indigenous people with high body fat, the risk of becoming pre-diabetics also increases the chance, but two-fold (OR = 2.18 [95% CI = 1.25 – 3.79]).

The unadjusted analysis also indicated the association of diabetes with age, schooling, paid work and any social benefits received. All the anthropometric variables were associated with diabetes among the indigenous people. It is worth noting that, among the Munduruku indigenous people who presented overweight (OR = 6.17 [95% CI = 2.60 – 4.64]) and obesity (OR = 61.03 [95% CI = 7.34 – 507.08]), the chances increased significantly. The clinical factors were also associated with diabetes, such as: pre-hypertension, hypertension and an increase in the total serum cholesterol level. On the other hand, the fact of having a Very Active level of physical activity (OR = 0.12 [95% CI = 0.02 – 0.68]) reduces by 88% the chance of the indigenous Munduruku developing diabetes.

Table 3 shows the Odds Ratio adjusted for gender and age of the variables that presented statistical significance (p≤0.05) in the analyses from Table 2, considering the two outcomes

**Table 2. Unadjusted odds ratio and Confidence Interval (CI) for sociodemographic and anthropometric variables, clinical factors, lifestyle and family history associated with pre-diabetes and diabetes among the Munduruku indigenous people.**

| Variables | Pre-Diabetes *vs* Normal Gross OR (95% CI) | p-value | Diabetes *vs* Normal Gross OR (95% CI) | p-value |
|---|---|---|---|---|
| *Sociodemographic Factors* | | | | |
| **Gender (Ref. Female)** | | | | |
| Male | 0.71 (0.40–1.25) | 0.235 | 0.63 (0.30–1.30) | 0.211 |
| **Age (years old)** | 1.04 (1.02–1.07) | **0.002** | 1.07 (1.04–1.11) | **<0.001** |
| **Marital Status (Ref. Has a partner)** | | | | |
| Without partner | 0.55 (0.32–0.96) | **0.037** | 0.60 (0.29–1.26) | 0.177 |
| **Schooling (Ref. Illiterate)** | | | | |
| Elementary School | 0.71 (0.20–2.54) | 0.593 | 0.32 (0.08–1.30) | 0.111 |
| High School | 0.43 (0.12–1.49) | 0.182 | 0.17 (0.04–0.70) | **0.014** |
| Higher Education or Postgraduate | 0.45 (0.11–1.77) | 0.253 | 0.22 (0.04–1.06) | 0.059 |
| **Paid work (Ref. Yes)** | | | | |
| No | 0.79 (0.45–1.39) | 0.420 | 0.42 (0.20–0.87) | **0.020** |
| **Social benefits (Ref. Yes)** | | | | |
| No | 1.35 (0.75–2.42) | 0.311 | 2.42 (1.15–5.07) | **0.019** |
| **Monthly family income (Ref. Does not have)** | | | | |
| <1 minimum wage (US$: 235.34) | 1.28 (0.25–6.45) | 0.768 | 1.40 (0.12–16.47) | 0.789 |
| 1–2 minimum wages (US$: 235.35–470.67) | 1.49 (0.29–7.67) | 0.631 | 2.18 (0.18–25.78) | 0.536 |
| 3–4 minimum wages (US$: 706.01–941.35) | 2.33 (0.39–13.91) | 0.353 | 3.67 (0.27–49.30) | 0.327 |
| ≥ 5 minimum wages (US$: 1,176) | 4.57 (0.35–59.12) | 0.245 | 2.00 (0.05–78.31) | 0.711 |
| *Anthropometric Factors* | | | | |
| **Neck circumference (cm)** | 1.06 (0.98–1.15) | 0.164 | 1.22 (1.09–1.37) | **<0.001** |
| **Waist circumference (cm)** | 1.07 (1.03–1.10) | **<0.001** | 1.14 (1.10–1.19) | **<0.001** |
| **Taper index** | 1.06 (1.02–1.09) | **0.002** | 1.14 (1.09–1.19) | **<0.001** |
| **BMI (kg/m$^2$)** | 1.20 (1.10–1.32) | **<0.001** | 1.38 (1.24–1.53) | **<0.001** |
| **BMI classification (Ref. Low weight/Normal weight** | | | | |
| Overweight | 1.87 (1.03–3.40) | **0.040** | 6.17 (2.60–14.64) | **<0.001** |
| Obesity | 12.76 (1.71–95.26) | **0.013** | 61.03 (7.34–507.08) | **<0.001** |
| **Body Fat Classification (Ref. Normal)** | | | | |
| Low | 1.96 (0.23–1.65) | 0.538 | - | - |
| High | 2.18 (1.25–3.79) | **0.006** | 10.19 (3.99–26.00) | **<0.001** |
| *Clinical Factors* | | | | |
| **Systolic blood pressure** | 1.02 (1.00–1.04) | 0.065 | 1.05 (1.02–1.08) | **<0.001** |
| **Diastolic blood pressure** | 1.04 (1.01–1.08) | 0.013 | 1.10 (1.05–1.15) | **<0.001** |
| **Blood Pressure Classification (Ref. Normal)** | | | | |
| Pre-hypertension | 2.41 (0.31–18.88) | 0.403 | 8.85 (1.03–76.36) | **0.047** |
| Hypertension | 3.11 (0.72–13.35) | 0.127 | 10.32 (2.22–47.92) | **0.003** |
| **Triglycerides** | 1.00 (1.00–1.01) | 0.122 | 1.01 (1.00–1.01) | <0.001 |
| **Total cholesterol** | 1.01 (1.00–1.02) | 0.222 | 1.02 (1.01–1.03) | **0.004** |
| *Habits and lifestyle* | | | | |
| **Smoker (Ref. No)** | | | | |
| Yes | 1.15 (0.67–1.99) | 0.616 | 1.48 (0.72–3.02) | 0.283 |
| **Physical activity level (Ref. Sedentary)** | | | | |
| Irregularly active | 0.54 (0.12–2.57) | 0.442 | 0.40 (0.07–2.22) | 0.292 |
| Active | 0.42 (0.09–1.89) | 0.258 | 0.33 (0.06–1.75) | 0.191 |

*(Continued)*

**Table 2.** (*Continued*)

| Variables | Pre-Diabetes *vs* Normal Gross OR (95% CI) | p-value | Diabetes *vs* Normal Gross OR (95% CI) | p-value |
|---|---|---|---|---|
| Very active | 0.31 (0.07–1.39) | 0.127 | 0.12 (0.02–0.68) | **0.016** |
| Consumption of Alcohol Beverages (Ref. Low risk consumption) | | | | |
| Risk intake, harmful or probable dependence | 1.10 (0.46–2.60) | 0.831 | 1.78 (0.49–6.43) | 0.378 |
| *Family History* | | | | |
| Hypertension (Ref. No) | | | | |
| Yes | 0.62 (0.28–1.39) | 0.250 | 1.14 (0.38–3.43) | 0.812 |
| Diabetes (Ref. No) | | | | |
| Yes | 0.93 (0.50–1.76) | 0.834 | 2.76 (1.06–7.19) | **0.038** |
| Stroke (Ref. No) | | | | |
| Yes | 0.56 (0.29–1.07) | 0.078 | 1.34 (0.55–3.24) | 0.515 |

Kwatá-Laranjal Indigenous Land, Borba, Amazonas, Brazil, 2018.

(pre-diabetes and diabetes). Thus, it is noteworthy that pre-diabetes was associated with increasing age, BMI and obesity. And diabetes remained associated with increasing age, BMI, overweight and obesity.

## Discussion

The prevalence of diabetes among the Munduruku indigenous people (12.2%) was higher than that found in other studies with indigenous populations, such as the Guarani, Kaiowá and Terena, from Dourados (Mato Grosso do Sul) (4.5%), Aymara, in Chile (1.5%) and was lower when compared to the Xavante indigenous people (25.9%) from Mato Grosso and to the Pima indigenous people from the state of Arizona (USA) [11, 16, 23, 24].

The largest participation in the study corresponded to the male gender (57.1%), unlike studies on cardiovascular risk carried out with other indigenous populations, such as: Xavante (49.2%) [18], Mura (42.2%) [25], Guarani-Kaiowá and Terena (44.2%) [11].

The mean age revealed that the Munduruku indigenous people were young adults: 36.6 years old (±14.7). A number of studies indicate that age is an important indicator for cardiovascular risk factors, especially for diabetes [18, 26, 27]. This study revealed that age presented a positive and significant association with the glycemic profile and, under this perspective, a study carried out with the Terena and Guarani indigenous peoples in 2016 also presented the same association [13].

**Table 3. Odds ratio adjusted for gender and age and confidence interval (CI) for sociodemographic and anthropometric variables, clinical factors, habits and life-style and family history associated with pre-diabetes and diabetes among the Munduruku indigenous people.**

| Variables | Pre-Diabetes *vs* Normal Adjusted OR (95% CI) | p-value | Diabetes *vs* Normal Adjusted OR (95% CI) | p-value |
|---|---|---|---|---|
| Age (years old) | 1.03 (1.00–1.06) | **0.032** | 1.05 (1.02–1.08) | **0.004** |
| BMI (kg/m$^2$) | 1.16 (1.06–1.28) | **0.002** | 1.28 (1.14–1.43) | **<0.001** |
| Overweight | 1.48 (0.79–2.77) | 0.226 | 4.07 (1.65–10.04) | **0.002** |
| Obesity | 9.26 (1.22–70.45) | **0.032** | 29.14 (3.38–251.04) | **0.002** |

Kwatá-Laranjal Indigenous Land, Borba, Amazonas, Brazil, 2018.

In relation to the socioeconomic conditions, the findings show a high proportion of low-income individuals: 46.6% with a family income of less than US$ 235.34, while 57.73% of the participants had no paid work and 61.66% were receiving social benefits from the Brazilian federal government. A study carried out with Mura de Autazes indigenous people (Amazonas) also revealed that 60.2% received income from some social benefits program of the Brazilian federal government and 59.4% had a family income of less than US$ 237.00 [25]. Another study carried out with the Guarani-Kaiowá and Terena indigenous peoples from Dourados (Mato Grosso do Sul) presented a percentage of 84.2% of families benefited by the *Bolsa Família* program, highlighting the conditions of social vulnerability experienced by the group and the possibility of social benefits improving the living conditions of the indigenous people [28]. In this context, it is worth noting that the Munduruku indigenous population presented risk for diabetes associated with low income.

The anthropometric data presented significant differences, revealing higher mean values among the indigenous people classified as diabetic compared to pre-diabetics and to those with normal blood glucose.

For the Body Mass Index, the global mean revealed excess weight [25.8 (±4.0) kg/m$^2$] among the Munduruku indigenous people, 38.3% of them with overweight and 14% with obesity. A study carried out in 2016 with the Mura de Autazes indigenous people (Amazonas), showed excess weight, with a BMI of 26.6 (±4.7) kg/m$^2$ [25]. Similar results were found among the indigenous women from the municipality of Dourados (Mato Grosso do Sul), who presented a mean BMI of 27.8 (±5.0) kg/m$^2$ [13]. When it comes to the Xavante Indigenous Reserves of São Marcos and Sangradouro, in the municipality of Volta Grande (Mato Grosso), the mean BMI indicates obesity among these indigenous people [30.3 (±5.1) kg/m$^2$] [18]. Overweight and obesity are worrisome conditions, as they increase the risk of developing cardiovascular diseases [18].

Among the Munduruku considered diabetic, the percentage of obesity was 28.3%. Flor et al. showed that, in 2008, the percentage attributable to obesity associated with diabetes mellitus was, for men, 37.3% in the Brazilian North Region against 45.4% in the entire country; and, for women, 55.1% in the North Region against 58.3% throughout Brazil, and the Brazilian mean was higher than the mean values found in the international literature [29].

When it comes to indigenous peoples, data for comparative analysis between diabetes and neck circumference are scarce. In our study, the mean neck circumference was 36 cm (±3.3), slightly below the national mean for the Brazilian male population (39.5±3.6) and slightly above the national mean for the Brazilian female population (34.0±2.9) [30]. In relation to other ethnic groups, such as Asian groups living in different cultural contexts, the mean found was 33 cm (±4.16), indicating that the increase in fat in the neck region had a greater indication of cardiometabolic disease when compared to the increase in the body and visceral mass index [31]. Another two studies conducted with the general American population suggest that increased neck circumference was associated with hypertension, diabetes, metabolic syndrome and dyslipidemia [32, 33].

In relation to waist circumference, the Munduruku presented a lower mean [85.3 cm (±10.2)], when compared to the Xavante indigenous people (Mato Grosso) [95.1 (±8.3) [34], but higher when compared to the Yanomami (Roraima) [76.3 (±46.8)] [35].

With regard to the Taper Index, the mean was 1.24 (± 0.9) among the Munduruku, similar to the one found among the Mura (municipality of Autazes, Amazonas) [1.27 (±0.08)] [25]. However, these findings are much lower when compared to the mean of the Brazilian population that varies between 1.35 (±0.08) and 1.34 (±0.09) [36, 37].

In relation to the blood pressure levels, the results of this study show that the prevalence of people with blood pressure levels suggestive of arterial hypertension was 10.2%. A systematic

review study with meta-analysis and meta-regression, conducted with indigenous people from the North Region (Ianomâmi, Suruí, Tembé, Amondaua, Parkatêjê, Suruí), from the Midwest Region (Terena, Zoró, Suyá, Kalapalo, Kuikuro, Matipus, Nahukwá, Mehináku, Waurá, Yawa-lapití, Guaraní, Tupinikin, Xavante, Khisêdjê and indigenous people from the Jaguapiru village), and from the Southeast and South Regions (Guaraní-Mbyá, Kaingang), showed a 12% increase in the chance of hypertension, in any indigenous person in Brazil, for each year studied [38]. The meta-analysis of this study showed that there was an increase in the prevalence of arterial hypertension, since in 1970 it was non-existent in the indigenous population, 0.1% (95% CI = 0.0% – 0.6%), when compared to 2014, when the highest prevalence of arterial hypertension was identified: 29.7% (95% CI = 26.1% – 44.4%) [38].

A study that investigated cardiovascular risk factors among different ethnic groups, living in the same urban area of Manaus (Amazonas), identified that, although the prevalence of SAH among the indigenous people was lower than in white-skinned (62.5%) and brown-/black-skinned (60.7%) individuals, that for pre-hypertension and hypertension was 28.6% among the Sateré-Mawé and 46.5% among ethnic groups from the upper Rio Negro [39].

During the assessment of the lipid levels, this study presented a mean of triglycerides of 165.5 (±86.5) mg/dL. In turn, 21.1% of the participants had high levels of triglycerides. These data are similar to those of the Mura de Autazes indigenous people (Amazonas) [163.5 (±104.7) mg/dL] [25] and Xavante of the São Marcos and Sangradouro Indigenous Reserves (Mato Grosso) [199.1 (±171.2) mg/dL] [18], differing from the mean among the Guarani-Mbyá indigenous people (Rio de Janeiro), which was 116.0 (±74.9) mg/dL [3].

Regarding the total cholesterol levels, the mean was 177.5 (± 32.2) mg/dL, considered within the boundary range and indicating that the Munduruku presented higher levels when compared to other ethnicities, such as the Sangradouro and the Guarani-Mbyá indigenous peoples, whose mean total cholesterol values were 145.8 (±4.7) mg/dL [16] and 143.8 (±28.8) mg/dL, respectively [3].

In relation to the diabetics indigenous individuals, 82.6% of them reported having a family history of diabetes and 52.8%, a family history of stroke. Indigenous people under the age of 55, who live in remote areas of Australia, were 14 times more likely to have an ischemic stroke, when compared to non-indigenous people belonging to the same age group. It is worth mentioning that the prevalence of diabetes found was 70.3% among indigenous people versus 34% among non-indigenous people [40].

With regard to the findings obtained through Odds Ratio adjusted for gender and age, it is possible to assert an increase in the chance of developing Pre-diabetes and Diabetes in relation to age in the group under study. Australian indigenous peoples had a 7% chance of developing diabetes each year of life [41]. A similar percentage was identified among the Munduruku, in which at each one year of life increase, there is a 3% chance of having pre-diabetes, and 5% for diabetes.

As for the BMI, for each increase in the unit of this ratio, the chance for the indigenous person becoming a pre-diabetic is 16%; and 28%, for diabetes. Data found in a comparative study between the population of the Aracruz Indigenous Reserve (Brazil) and the population of Espírito Santo (Brazil) showed that obese non-indigenous men and women were twice as likely to have DM but, when it comes to the indigenous people in this study, no significant differences were found [42].

Among the overweight indigenous people, the chance of having diabetes is four times higher, respectively. For the obese, on the other hand, the chances substantially increase, both for pre-diabetes, which increases to nine times, and for diabetes, which can reach 29 times. In the study with Guarani, Kaiowá and Terena, from the Jaguapiru village (Mato Grosso do Sul), the prevalence of diabetes among women was 9% and among men, 5%. The study also

indicated a positive and significant association between obesity and diabetes (PR = 1.88; 95% CI = 1.45 – 2.43; p<0.001). A population-based study carried out in different Brazilian regions showed that obese individuals had 35% [95% CI = 1.35 – 1.86; p<0.001] chances of developing diabetes [43]. These findings show that the Munduruku, although still distant from the national mean of the general Brazilian population, are in an unfavorable condition toward the development of diabetes in relation to other ethnic groups living in a similar cultural context.

### Study limitations

In the absence of specific cut-off points for indigenous populations, those used for the general population were considered, also adopted in other studies on different ethnic groups.

As it was impossible to apply a dietary recall, it was not possible to verify how much the eating habits are associated with the values found for glucose, cholesterol and triglycerides.

The instruments adopted in the interview were not specific to indigenous peoples. However, since it is an essential requirement to achieve the proposed objectives, the adequacy of language to the understanding of the group under study constituted a task that demanded different moments of planning and evaluation by the team.

### Conclusion

The 12% prevalence of glycaemia found among the Munduruku indigenous people is suggestive of diabetes mellitus, and that of 74.3%, revealing pre-diabetes, configure themselves as worrying indexes, as well as the chance of pre-diabetes, which increases by 20% when the BMI increases by one unit. It is necessary to consider changes in the eating habits and lifestyle, as well as environmental and social changes that can affect the health of the Munduruku, and consider the stress levels, with the possibility of each of these elements contributing or not to the results of this study. Consequently, it becomes indispensable to develop strategies combining early diagnosis and treatment actions with actions to reduce the risk factors, in order to meet the needs and singularities of the Munduruku indigenous people. It is also suggested to develop new research studies on the topic in order to consolidate these findings in other Munduruku indigenous contexts.

### Author Contributions

**Conceptualization:** Zilmar Augusto de Souza Filho, Noeli das Neves Toledo, António Manuel Sousa, Gilsirene Scantelbury de Almeida.

**Data curation:** Hanna Lorena Moraes Gomes, Neuliane Melo Sombra.

**Formal analysis:** António Manuel Sousa.

**Funding acquisition:** Noeli das Neves Toledo.

**Investigation:** Hanna Lorena Moraes Gomes, Neuliane Melo Sombra.

**Methodology:** Hanna Lorena Moraes Gomes, Neuliane Melo Sombra.

**Project administration:** Zilmar Augusto de Souza Filho, Noeli das Neves Toledo, Gilsirene Scantelbury de Almeida.

**Supervision:** Zilmar Augusto de Souza Filho, Evelyne Marie Therese Mainbourg, Gilsirene Scantelbury de Almeida.

**Visualization:** Hanna Lorena Moraes Gomes, Eliza Dayanne de Oliveira Cordeiro, Zilmar Augusto de Souza Filho, Evelyne Marie Therese Mainbourg, Gilsirene Scantelbury de Almeida.

**Writing – original draft:** Hanna Lorena Moraes Gomes, Neuliane Melo Sombra, Eliza Dayanne de Oliveira Cordeiro.

**Writing – review & editing:** Hanna Lorena Moraes Gomes, Neuliane Melo Sombra, Eliza Dayanne de Oliveira Cordeiro, Evelyne Marie Therese Mainbourg, Gilsirene Scantelbury de Almeida.

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
