## [Decision Letter · Decision Letter 0]

5 Mar 2021

PONE-D-21-00872

Glycemic profile and associated factors in Munduruku indigenous people, Amazonas.

PLOS ONE

Dear Dr. Moraes Gomes,

Thank you for submitting your manuscript to PLOS ONE. After careful consideration, we feel that it has merit but does not fully meet PLOS ONE’s publication criteria as it currently stands. Therefore, we invite you to submit a revised version of the manuscript that addresses the points raised during the review process.

An expert in the field and myself has reviewed your manuscript. Some important concerns should be clarified in order to consider the manuscript for publication.

We look forward to receiving your revised manuscript.

Kind regards,

Fernando Guerrero-Romero, MD, PhD

Academic Editor

PLOS ONE

Journal Requirements:

2.We note you have included a table to which you do not refer in the text of your manuscript. Please ensure that you refer to Table 3 in your text; if accepted, production will need this reference to link the reader to the Table.

Additional Editor Comments:

Although authors state how sample size was calculated, the sampling strategy must be clearly stated

In the Abstract section authors state that they performed a casual measurement of fasting capillary glycemia, which is confusing. Was a casual measurement? Or Was a fasting measurement?, please clarify.

Which were the automatic devices used for measurement of glycemia and lipid profile?

Which were the inter- and inter-assay variations of the glucose and lipid profile measurements?

Which kind of device was used for measurement of blood pressure?

How was standardized the inter-observer variations for anthropometric measurements?

How can be explained the elevated prevalence of pre-diabetes (74.3%).

According data in the Results section, whereas prevalence of overweight plus obesity was 52.7%, prevalence of altered glucose levels was 86.5%. Also it draws attention the low prevalence of hypertension. These findings must be extensively discussed.

Numerical data for anthropometric and biochemical measurements are mandatories.

How can influence in the results and conclusions using blood capillary glucose? In addition to recognize this as a limitation, also should be discussed as a potential source of bias.

Reviewers' comments:

Reviewer's Responses to Questions

**Comments to the Author**

1. Is the manuscript technically sound, and do the data support the conclusions?

Reviewer #1: Partly

2. Has the statistical analysis been performed appropriately and rigorously? 

Reviewer #1: Yes

3. Have the authors made all data underlying the findings in their manuscript fully available?

Reviewer #1: No

4. Is the manuscript presented in an intelligible fashion and written in standard English?

Reviewer #1: No

5. Review Comments to the Author

Reviewer #1: The goal of this study was to characterize the prevalence of impaired glucose metabolism among the Munduruku indigenous people in the Brazilian Amazon. This article's aim is needed and merit-worthy. I applaud the authors for their interest in addressing this topic among populations that have been historically excluded from major health assessments, and who are also undergoing profound transformations in their nutritional and epidemiological patterns. However, this manuscript has several critical flaws; thus, I suggest this article be accepted only if the major revisions suggested below are fully addressed. My main concerns relate to the many limitations of this study and the lack of acknowledgment of the issues arising from their limitations. In particular, it is not clear in the methods how they ensured that participants actually fasted for <8 hrs before data collection. This is a pressing issue for this study, especially since the prevalence of "prediabetes" seems surprisingly high, and not even close to the prevalence observed at any other population in the region at the least. Another major issue throughout this article is the many assumptions and associations the authors establish without providing evidence of causality or even references for other populations/studies. The most pressing revisions to be made include backing their claims with robust evidence from other studies, and more attention to the many issues in the limitations and methods sections in particular - as in many occasions, it is not clear the instruments used to collect the data. Minor but still essential issues include a better definition and operationalization of some concepts used, further clarification of some ideas, and proofreading.

As neither the Pdf nor the Word document had page numbers or page lines, all comments indicated below refer to the pdf proof page numbers (PONE-D 21 00872_reviewer.pdf). Please consider including both page and line numbers in your file (s) in further submissions to make revisions easier to refer to. In addition, the authors indicate that "all relevant data are within the manuscript" but data is not available (as supplementary material or in a repository). Only descriptive statistics are available.

ABSTRACT:

P.7. Spelling mistake: A cross-sectional

P.7. (but also throughout the manuscript). If it's fasting glucose, then it is not casual/random

INTRODUCTION:

P. 8 and 9 and other pages throughout the article. Clarify how the authors understand "vulnerability"/"being vulnerable"/"condition of vulnerability" in this context. Be more specific about what vulnerabilities is this population potentially exposed to that could have an effect on the topic studied.

P. 8. Quote defining DM should be followed by a reference and page number, if this is an exact quote.

P.9. Authors indicate that "indigenous populations seem to be susceptible to this disease" (DM); however, they do not provide any references supporting this claim. In addition, if the authors believe that this could be an important factor explaining the high prevalence in this (and other?) indigenous groups, they should address this further in the discussion section.

P.9. "Midwest region." Consider indicating Brazilian Midwest or name the region for international readers.

METHODOLOGY

P.9. The entire sentence of this section is incomplete. Rephrase this.

STUDY PARTICIPANTS

I suggest including a map of the location of the villages sampled in the region. Most international readers are probably unaware of the geographical context where this study was carried out.

P.10. Word choice. Consider replacing "was formed" with "comprised" or similar.

P.10. The authors indicate that the sample excluded "those who had difficulty

communicating in Portuguese." I wonder whether it is possible that by excluding these individuals, the researchers are also excluding those who are more likely to follow 'traditional' diets or lifestyles. I suggest discussing this in the limitations if this might be the case in this context.

DATA COLLECTION

- How was physical activity assessed? Were people asked whether they were sedentary or physically active directly? Or did the researchers used standardized physical activity questionnaires adapted to the local context? If so, indicate the questionnaire used and how this data was converted to physical activity categories. On the other hand, if authors assessed physical activity through accelerometers, this should be clearly indicated as well.

- How was alcohol consumption estimated? If this was assessed through questionnaires, this should be indicated - and also the time frame covered by this questionnaire. In general, provide more detail in the text or supplementary materials about the questionaries and categories used.

There are several limitations when assessing this information through questionnaires; hence, this should be discussed in the limitations section.

P.10. What do the authors mean by physical examination? Please clarify.

P.10. is this waist circumference? Indicate in this section the methodology followed to perform these measurements, especially for neck and waist circumference.

P.10. How did the researchers ensure that participants did not have any sweetened beverages or even a snack for eight hours before data collection? This seems tricky in a context like this, and given the exceedingly high prevalence of prediabetes reported, it might be the case that several participants had eaten/drunk something in the previous 8hrs. This is a critical issue, so I am wondering whether the researchers confirmed (asked?) the participants if they ate/drunk in the previous 8 hrs at least? If not, this should definitely be addressed in the limitations section.

P.10 and throughout the manuscript, I would suggest to avoid referring to the participants as "indigenous people" in every instance. Authors could use "participants" or "individuals" in some parts of the article.

P.10. Authors indicate that "alterations in capillary glycemia and/or any of the tests performed were referred to …" Clarify here for the case of other biomarkers that were considered "altered". Did the researchers referred people classified as 'obese' to the health post as well? Please, be more specific. Also, indicate the specific cut-offs and references used to classify 'altered' versus normal somewhere before this paragraph (or this paragraph could go after the authors indicate the cut-offs used).

P.11. "automatic calibrated device." What exactly is this device? Is it a portable blood analyzer such as Cardiocheck? Indicate details about the device and strips used to measure, if applicable.

P.11. BMI categories indicated do not include individuals with a BMI < 18.5? I assume none of the individuals have a BMI < 18.5, or where this excluded from the analysis? Either way, this should be indicated in the text.

P.11. Clarify what casual blood pressure measurement means in this assessment. Did the authors take just one measurement or more, and they averaged them? In addition, indicate what "automatic calibrated device" was used to perform these measurements.

P.11. What do the authors mean by "The preparation of the indigenous people"? please clarify or delete if unnecessary.

STATISTICAL ANALYSIS

Indicate in this section which variables were considered explicitly as independent (predictors) and dependent (outcome) and covariates, if any.

P.12. It is unclear what "variables were compared using hypothesis tests" mean. Consider indicating the specific hypothesis being tested and the statistical tests used.

P.12. A couple of sentences in this paragraph are grammatically odd or incorrect. Consider further checking English grammar or syntax

P.12. The authors indicate that "collinearities and less relevant variables were eliminated." Which variables were the ones eliminated? How the authors define "less relevant" variables?

RESULTS

Refer to Tables 1 and 2 when describing the results.

P.13. How do the authors know about the association between education level and wage labor in this population? Is there any data/study in these populations - or others in the region - they can refer to where these variables are correlated? There is no clear link between cause and effect to indicate "consequence" here.

P.13. Do the authors mean understandable as expected? Please consider a different word choice.

P.13. Indicate what the socioeconomic classification (D, E) means for international readers.

P.13 "Regarding body fat, the increase was high (28.3%) and very high (33.3%)." Please clarify what they mean by "the increase was high."

P.14 Throughout the table, be consistent in the use of %. Some categories have the % symbols, and others do not. It does not matter which the authors choose, but please be consistent.

p.16. The authors say that "(…) more likely to become pre-diabetic". This is a cross-sectional study; thus, the authors cannot really test for causality, and therefore cannot assert that X individuals were more likely to become Y. Consider phrasing this as associations or as direct observations.

P.17-18. Table 2. The "1" throughout the table as the reference seems unnecessary. The authors could indicate the reference in the variable name or somewhere in the methods.

P.18. Authors say that "… selected only the variables with statistical significance presented in Table 2.". Do they mean that they include only variables that were significantly different between diabetes categories, or other categories? Please clarify in the text.

P.19. Please rephrase the following sentence as it is not clear what the authors mean: "... where there was prevalence of women: Xavante (50.8%)(22), Mura (57.8%)(23), Guarani, Kaiowá and Terena (55.8%)."

DISCUSSION:

P.20. The authors claim that income explains ultraprocessed/industrialized food consumption. Some evidence should be provided to make this claim. If they did not collect dietary data, at least they could refer to other studies in the region where such an association (between ultraprocessed foods and diabetes) has been found and discuss how this might explain some of the results found.

P.21. Do the authors mean low frequencies of physical activity categorize as active/very active? Please clarify.

P.22. The authors mention the prevalence of diabetes observed in other studies among indigenous populations in the Americas, but they do not provide the references for these studies. This happens in many other parts of the text, and on several other occasions, the references are not included in the first mention of the study discussed. Please check that references are appropriately included throughout the text.

P.22. In the sentence "The causes that justify these frequencies are not fully clarified." Do the authors mean the causes that explain the frequencies? Consider a different word choice.

P.22. The references for the claim that a number of studies indicate a high prevalence of obesity should be right after this sentence, and not at the end of the paragraph

P.22. The authors assert that excessive salt and sugar consumption are risk factors for cardiovascular diseases but do not cite any literature to back this claim.

LIMITATIONS:

P.23. I fully agree with the authors that it may be that the standard reference values for CVD or diabetes risk categories might not be the actual risk values for the population studied; however, if the authors consider this to be a limitation of their study, they should expand this discussion and cite other literature that has addressed this issue. There is plenty of literature on different cut-offs for Asian populations vs. the standard White-European population commonly used in research.

P.23. The authors indicate that the results can be less accurate than those of measurements using blood samples. Do they mean "venous" blood samples? Please clarify.

P.23. The authors emphasized stress as a likely contributing factor to the high prevalence of DM. If they consider that this variable is a relevant one in the outcome, they should further address this issue in the discussion, citing appropriate literature on the topic

6. PLOS authors have the option to publish the peer review history of their article (what does this mean?). If published, this will include your full peer review and any attached files.

Reviewer #1: **Yes: **Catalina I. Fernández

---

## [Author Response · Author response to Decision Letter 0]

30 May 2021

We would like to thank you for all the suggestions and clarify that the topics methods, statistical analysis, ethical aspects, results, discussion and limitations of the study have been revised and have been rewritten to meet the reviewers' suggestions.

We also inform that some variables have been inserted in the tables:

Monthly family income (minimum salary)

Neck circumference (cm), mean (SD)

Waist circumference (cm), mean (SD)

Conicity index, mean (SD)

BMI (kg / m2), mean (SD)

BMI classification

Classification of body fat

Systolic blood pressure SBP (mmHg), mean (SD)

Diastolic blood pressure DBP (mmHg), mean (SD)

Response to reviewers:

We note that you have included a table that you do not refer to in the text of your manuscript. Make sure to refer to Table 3 in your text; if accepted, the production will need this reference to link the reader to the Table: Adjustment made. Thanks for listening.

Additional editor comments:

Although the authors state how the sample size was calculated, the sampling strategy must be clearly indicated: The text of the Summary section has been reworked to meet demand. Grateful for the attention.

In the Summary section, the authors state that they performed a casual measurement of fasting capillary blood glucose, which is confusing. Was it a casual measurement? Or was it a fast measurement? Please clarify: Excluded the word “casual” that had been used incorrectly.

What were the automatic devices used to measure blood glucose and lipid profile? Portable digital device from Accu-Check® Active for measuring capillary blood glucose and the Accutrend® Plus device for measuring cholesterol and triglycerides. More information is in the text. Grateful for the attention.

What were the inter- and inter-assay variations of glucose and lipid profile measurements? Sorry, but we don't understand this question.

What type of device was used to measure blood pressure? Apparatus brand OMRON / Model HBP-1100. More information is in the text. Grateful for the attention.

How were interobserver variations standardized for anthropometric measurements? Anthropometric measurements were taken by the same observer in all participating individuals, in order to standardize these data.

As explained by the high prevalence of pre-diabetes (74.3%). According to the data in the Results section, while the prevalence of overweight plus obesity was 52.7%, the prevalence of altered blood glucose was 86.5%. It also draws attention to the low prevalence of hypertension. These findings should be widely discussed: Suggestion accepted, thank you very much.

Numerical data for anthropometric and biochemical measurements are mandatory: Data has been inserted in the text

How can it influence the results and conclusions of the use of capillary glycemia? In addition to recognizing this as a limitation, it should also be discussed as a potential source of bias: Done

Making data public: The data used in this research were collected in indigenous populations, protected by specific laws to these populations in Brazil. That way, we cannot make this data available. According to Brazilian legislation (http://www.planalto.gov.br/ccivil_03/_ato2011-2014/2011/lei/l12527.htm), the authors are responsible for the confidentiality and confidentiality of the analyzed data. Therefore, free access to the data used in our research is not allowed. If some researcher needs to access the data set, can request it from the Ethics Committee of the federal university of Amazonas, where the macro project is registered.

Details for contact:

Project Protocol: CAAE UFAM No 74361617.2.0000.5020

Research Ethics Committee of the Nursing School of Manaus

Rua Terezina, 495 - Adrianópolis, Manaus - AM, CEP: 69057-070

Email: cep@ufam.edu.br

Improve English grammar: Thanks for the suggestion

Ensure in the method that the participants were fasting for 8 hours: Clarified in the text. Thanks.

Clarify the instrument used in data collection: Suggestion answered.

Put page number and line in the document: Done

Reviewers' comments:

ABSTRACT:

P.7. Spelling error: a cross section: Done, the word “section” is added before “cross section”.

P.7. (but also throughout the manuscript). If it is fasting glucose, then it is not casual / random: Expressions taken from the text.

INTRODUCTION:

P. 8 and 9 and other pages throughout the article. Clarify how the authors understand "vulnerability" / "being vulnerable" / "condition of vulnerability" in this context. Be more specific about what vulnerabilities this population is potentially exposed to that may affect the topic studied: The term vulnerability has been clarified in the text.

P. 8. Quote that defines DM should be followed by a reference and page number, if this is an exact quote: Done

P.9. Authors indicate that "indigenous populations seem to be susceptible to this disease" (DM); however, they do not provide any reference after.

ie that statement. In addition, if the authors believe that this may be an important factor in explaining the high prevalence in these (and others?) Indigenous groups, they should address this further in the discussion section: Reformulated text in the manuscript

P.9. "Midwest region." Consider indicating the Brazilian Midwest or name the region for international readers: Done.

METHODOLOGY

P.9. The entire sentence in this section is incomplete. Rephrase that: Rephrased session. Grateful for the attention.

STUDY PARTICIPANTS

I suggest including a map of the location of the sampled villages in the region. Most international readers are probably unaware of the geographic context in which this study was conducted: Suggestion accepted.

P.10. Choice of words. Consider replacing "was formed" with "understood" or similar: Done

P.10. The authors indicate that the sample excluded "those who had difficulty communicating in Portuguese." I wonder if it is possible that, in excluding these individuals, researchers also exclude those who are more likely to follow 'traditional' diets or lifestyles. I suggest discussing this in the limitations, if applicable in this context: Text was reformulated, as there was no participant who did not speak Portuguese.

DATA COLLECTION

- How was physical activity evaluated? Were people asked if they were sedentary or physically active directly? Or did the researchers use standardized physical activity questionnaires adapted to the local context? If so, please indicate the questionnaire used and how this data was converted into categories of physical activity. On the other hand, if the authors evaluated physical activity using accelerometers, this should also be clearly indicated: We evaluated using a standardized and validated questionnaire called the International Physical Activity Questionnary (IPAQ). More information is inserted in the text.

- How was alcohol consumption estimated? If this was assessed using questionnaires, it must be stated - and also the time period covered by this questionnaire. In general, provide more details in the text or complementary materials on the questionnaires and categories used: Alcohol consumption was also through a validated questionnaire called Alcohol Use Disorder Identification Test (AUDIT), more details are in the text of the document.

There are several limitations when evaluating this information through questionnaires; therefore, this should be discussed in the limitations section: Suggestion accepted.

P.10. What do the authors mean by physical examination? Please clarify: Expression taken from the text. Redesigned text.

P.10. Is that waist circumference? Indicate in this section the methodology followed to perform these measurements, mainly for neck and waist circumference: Information clarified in the text.

P.10. How did the researchers ensure that the participants did not eat sugary drinks or even a snack for eight hours before data collection? This seems complicated in a context like this and, given the extremely high prevalence of reported pre-diabetes, it may be the case that several participants have eaten / drank something in the previous 8 hours. This is a critical question, so I would like to know if the researchers confirmed (asked?) To the participants if they ate / drank in the previous 8 hours, at least. Otherwise, this should definitely be addressed in the limitations section: Text reworked in order to clarify the questioning.

P.10 and throughout the manuscript, I suggest avoiding referring to participants as "indigenous peoples" in all cases. Authors can use "participants" or "individuals" in some parts of the article: Suggestion accepted.

P.10. Authors indicate that “changes in capillary glycemia and / or any of the tests performed were referred to…” Clarify here for the case of other biomarkers that were considered “altered”. Did the researchers also refer people classified as 'obese' to the health center? Please be more specific. Also, indicate the specific cutoff points and references used to classify 'changed' versus normal somewhere before this paragraph (or this paragraph can go after the authors indicate the cutoff points used): Adjusted text. Thanks for the collaboration.

P.11. "device automatically calibrated." What exactly is this device? Is it a portable blood analyzer like Cardiocheck? Indicate details about the device and strips used to measure, if applicable: Complete information has been inserted in the text.

P.11. Do the indicated BMI categories not include individuals with a BMI <18.5? I assume that none of the individuals has a BMI <18.5, or has it been excluded from the analysis? In any case, this must be indicated in the text: Corrected information in the text and in the table. BMI <18.5 was considered.

P.11. Clarify what casual blood pressure measurement means ica in this assessment. Did the authors take just one measure or more and average them? In addition, indicate which "automatically calibrated device" was used to perform these measurements: "Casual" expression was removed from the text because it was causing misinterpretation. And a translation error occurred. It is not automatically calibrated, it is an automatic and calibrated device, complete information in the text.

P.11. What do the authors mean by "The preparation of the indigenous"? Clarify or delete if not necessary: reformulated text, the expression has been removed from the text.

STATISTICAL ANALYSIS

This topic has been rewritten to better answer the editors' questions.

Indicate in this section which variables were explicitly considered to be independent (predictors) and dependent (result) and covariates, if any.

P.12. It is not clear what "variables were compared using hypothesis tests" means. Consider indicating the specific hypothesis being tested and the statistical tests used.

P.12. Some sentences in this paragraph are grammatically strange or incorrect. Consider checking English grammar or syntax: suggestion accepted.

P.12. The authors indicate that “the collinearities and less relevant variables have been eliminated”. What variables have been eliminated? How do the authors define "less relevant" variables?

RESULTS

Refer to Tables 1 and 2 when describing the results.

The entire topic of results has been rewritten to better answer the reviewers' questions

P.13. How do the authors know about the association between education level and salaried work in this population? Is there any data / study in these populations - or in others in the region - that can refer to where these variables are correlated? There is no clear link between cause and effect to indicate "consequence" here.

P.13. Do the authors mean understandable as expected? Please consider a different word choice: reformulated text.

P.13. Indicate what the socioeconomic classification (D, E) means for international readers: reformulated text.

P.13 "Regarding body fat, the increase was high (28.3%) and very high (33.3%)." Please clarify what they mean by "the increase was high": reformulated text.

P.14 Throughout the table, be consistent in using%. Some categories have% symbols and others do not. It doesn't matter the choice of authors, but be consistent: Done

p.16. The authors state that "(...) more likely to become pre-diabetic". This is a cross-sectional study; therefore, authors cannot really test causality and, therefore, cannot claim that X individuals were more likely to become Y. Consider formulating this as associations or as direct observations: Recast text.

P.17-18. Table 2. The "1" in the entire table as a reference seems unnecessary. Authors can indicate the reference in the variable name or somewhere in the methods: Corrected table.

P.18. Authors state that "... selected only the variables with statistical significance presented in Table 2.". Do they mean that they include only variables that were significantly different between the categories of diabetes or other categories? Please clarify in the text: Recast text.

P.19. Rephrase the following sentence, as it is not clear what the authors mean: "... where there was a predominance of women: Xavante (50.8%) (22), Mura (57.8%) (23), Guarani, Kaiowá and Terena (55.8%). ": Recast text.

DISCUSSION:

This session was rewritten to better respond to the reviewers' suggestions.

P.20. The authors state that income explains the consumption of ultra-processed / processed foods. Some evidence must be provided to make this claim. If they did not collect dietary data, at least they could refer to other studies in the region where such an association (between ultra-processed foods and diabetes) was found and discuss how it could explain some of the results found.

P.21. Do the authors consider low frequencies of physical activity categorized as active / very active? Please clarify.

P.22. The authors mention the prevalence of diabetes seen in other studies among indigenous populations in the Americas, but do not provide references for these studies. This happens in many other parts of the text and, on several other occasions, references are not included in the first mention of the study discussed. Check that references are included properly throughout the text.

P.22. In the sentence “The causes that justify these frequencies are not fully clarified”. Do the authors refer to the causes that explain the frequencies? Consider a different word choice.

P.22. References for the statement that a series of studies indicate a high prevalence of obesity should be right after this sentence, and not at the end of the paragraph

P.22. The authors claim that excessive consumption of salt and

sugar are risk factors for cardiovascular disease, but they do not cite any literature to support this claim.

LIMITATIONS:

P.23. I fully agree with the authors that it may be that the standard reference values for CVD or diabetes risk categories are not the actual risk values for the population studied; however, if the authors consider this to be a limitation of their study, they should expand this discussion and cite other literature that has addressed this issue. There is a lot of literature on different cutoff points for Asian populations versus the standard white European population commonly used in research.

P.23. The authors indicate that the results may be less accurate than measurements with blood samples. Do they mean "venous" blood samples? Please clarify: Expression taken from the text.

P.23. The authors emphasized stress as a likely contributing factor to the high prevalence of DM. If they consider that this variable is relevant in the result, they should address this issue in more detail in the discussion, citing the appropriate literature on the subject: Expression taken from the text.

---

## [Editor Report · Decision Letter 1]

11 Jun 2021

PONE-D-21-00872R1

Glycemic profile and associated factors in Munduruku indigenous people, Amazonas.

PLOS ONE

Dear Dr. Hanna Lorena Moraes Gomes,

Thank you for submitting your manuscript to PLOS ONE. After careful consideration, we feel that it has merit but does not fully meet PLOS ONE’s publication criteria as it currently stands. Therefore, we invite you to submit a revised version of the manuscript that addresses the points raised during the review process.

Authors have responded practically all the raised concerns improving the manuscript. However, in order to appropriately evaluate changes made, it is important that all changes made through manuscript in response to concerns must be highlighted. The response to reviewers also must indicate the page, paragraph, and lines were the changes performed can be located in the text.

We look forward to receiving your revised manuscript.

Kind regards,

Fernando Guerrero-Romero, MD, PhD

Academic Editor

PLOS ONE

Journal Requirements:

Additional Editor Comments (if provided):

Authors have responded practically all the raised concerns improving the manuscript. However, in order to appropriately evaluate changes made, it is important that all changes made through manuscript in response to concerns must be highlighted. The response to reviewers also must indicate the page, paragraph, and lines were the changes performed can be located in the text.

---

## [Author Response · Author response to Decision Letter 1]

16 Jul 2021

Response to reviewers

We would like to thank you for all your suggestions and clarify that the topics of methods, statistical analysis, ethical aspects, results, discussion and limitations of the study underwent a review and were rewritten to meet the reviewers' suggestions.

We also inform that the variables below were inserted in tables 1 and 2 and are highlighted in yellow in the text.

Monthly family income (minimum salary)

Circunferência do pescoço (cm), mean (SD)

Circunferência da cintura (cm), mean (SD)

Conicity index, mean (SD)

IMC (kg/m2), mean (SD)

Classificação do IMC

Classificação da gordura corporal

Systolic blood pressure SBP (mmHg), mean (SD)

Diastolic blood pressure DBP (mmHg), mean (SD)

Response to reviewers:

We note that you have included a table that you do not reference in your manuscript text. Be sure to refer to Table 3 in your text; if accepted, the production will need this reference to link the reader to the Table: Table 3 was mentioned in line 267 at the beginning of the paragraph in response to the reviewers' request.

Additional Editor Comments:

Although the authors state how the sample size was calculated, the sampling strategy must be clearly stated: The text of the Summary section, in line 15, has been re-written to meet demand. Grateful for the attention.

In the Summary section, the authors claim that they performed a casual measurement of fasting capillary blood glucose, which is confusing. Was it a casual measurement? Or was it a measurement of fasting?, please clarify: Deleted the word “casual” that had been misused in the summary section in line 15.

What were the automatic devices used to measure blood glucose and lipid profile? Portable digital device from the manufacturer Accu-Check® Active for measuring capillary blood glucose and the Accutrend® Plus device for measuring cholesterol and triglycerides. More information is in the text. Thank you for your attention, indicated in the text in lines 129 through 132.

What were the inter- and inter-assay variations in glucose and lipid profile measurements? Sorry, but we don't understand this question.

What type of device was used to measure blood pressure? OMRON branded/Model HBP-1100 device. More information is in the yellow text described in line 155. Thanks for your attention.

How were interobserver variations standardized for anthropometric measurements? For the data collection step, the team received specific training in order to standardize the procedures for: measurement of blood glucose and capillary lipid levels after fasting for at least eight hours, measurement of blood pressure levels, measurements of anthropometric measurements and conducting the interview, this information is described in the text, underlined in yellow, in lines 114 to 117.

This explains the high prevalence of pre-diabetes (74.3%). According to the data in the Results section, while the prevalence of overweight plus obesity was 52.7%, the prevalence of altered blood glucose was 86.5%. The low prevalence of hypertension is also noteworthy. These findings should be widely discussed: Suggestion accepted, discussion regarding obesity described in lines 320 through 324; Regarding blood pressure levels, data are described in lines 342 to 354. Regarding data on diabetes and altered blood glucose, information is available in lines 281 to 285.

Numerical data for anthropometric and biochemical measurements are mandatory: The cutoff point used for assessment and classification of fasting capillary blood glucose was: normal < 100 mg/dL, prediabetes: de 100 and < 126 mg/dL and diabetes : ≥ 126 mg/dL. For lipid levels, the classification was: hypercholesterolemia when ≥ 240 mg/dL and hypertriglyceridemia when ≥ 175 mg/dL, available in lines 132 to 136. The cutoff points adopted to classify neck circumference measurements were: ≥ 37 cm for men and ≥ 34 cm for women; and waist circumference were: ≥ 102 cm for men and ≥ 88 cm for women. For the conicity index, the following values were adopted: ≥ 1.25 for men and ≥ 1.18 for women. As for the body mass index (BMI) it was classified as: underweight (<18.5kg/m2), normal weight (18.5-24.9 kg/m2), overweight (from 25.0 to 29.9 kg/m2) and obesity (≥30 ,0 kg/m2)(21). The classification of the percentage of body fat considered the stratification by age and gender: Low (<8.0-<13.0% for men and <21.0-<30.0% for women), Normal (13.0-24.9% for men and 30.0% -≤35.9% for women) and High (≥25.0% for men and ≥36.0% for women), described in lines 145 to 153.

How can it influence the results and conclusions of the use of capillary blood glucose? In addition to recognizing this as a limitation, it should also be discussed as a potential source of bias: Discussed in text on lines 407 through 410.

Make the data public: The data used this research were collected in indigenous populations, protected by specific laws to these populations in Brazil. That way, we cannot make this data available. According to Brazilian legislations (http://www/planalto.gov.br/ccivil_03/_ato2011-2014/2011/lei/ll2527.htm), the authors are responsible for the confidentiality and confidentiality of the analyzed data. Therefore, free access to the data used in our research is not allowed. If somo researcher needs to access the data set, can request it from the Ethics Committee of the Federal University of Amazonas, where the macro project is registered.

Details for contact:

Project Protocol: CAAE UFAM Nº 74361617.2.0000.5020

Manaus School of Nursing Research Ethics 

Address: Road Terezina, N. 495 – Adrianópolis, Manaus – Amazonas – Brazil 

CEP: 69057-070

E-mail: cep@ufam.edu.br

Improve English grammar: Grammar proofreading was performed to meet reviewers' recommendations. Thanks for the suggestion

Ensure in the method that participants were fasting for 8 hours: Clarified in the text, underlined in yellow, in lines 120 to 123. Thank you.

Clarify the instrument used in data collection: Suggestion answered, information described in the text in lines 173 to 177.

Put page and line number in the document: Done

Reviewers' comments:

ABSTRACT:

P.7. Spelling error: a cross-section: Done, added the word “cut” before “cross” in the text in line 12.

P.7. (but also throughout the manuscript). If it's fasting glucose, then it's not casual/random: Expressions taken from the text.

INTRODUCTION:

P. 8 and 9 and other pages throughout the article. Clarify how the authors understand "vulnerability" / "being vulnerable" / "condition of vulnerability" in this context. Be more specific about which vulnerabilities this population is potentially exposed to that can affect the topic studied: The term vulnerability was described in the text in lines 42 to 47.

Q. 8. Citation defining DM should be followed by a reference and page number if this is an exact quote: Done, info is on line 59.

P.9. Authors indicate that "indigenous populations seem to be susceptible to this disease" (DM); however, they do not provide any references to support this claim. Furthermore, if the authors believe that this may be an important factor in explaining the high prevalence in this (and other?) Indigenous groups, they should address this further in the discussion section: Text reworded in manuscript in lines 76 through 79.

P.9. "Midwest region." Consider indicating the Brazilian Midwest or name the region for international readers: Information entered in line 80.

METHODOLOGY

P.9. The entire sentence in this section is incomplete. Rephrase this: Recast Session, lines 83 through 183. Grateful for your attention.

STUDY PARTICIPANTS

I suggest including a map of the location of the sampled villages in the region. Most international readers are probably unaware of the geographic context in which this study was conducted: Suggestion accepted, including information about the figure in line 93.

P.10. Choice of words. Consider replacing "was formed" with "understood" or similar: Reworded text.

P.10. The authors indicate that the sample excluded “those who had difficulty communicating in Portuguese.” I wonder if it is possible that, by excluding these individuals, researchers also exclude those who are more likely to follow 'traditional' diets or lifestyles. I suggest discussing this in the limitations, if applicable in this context: Text was reformulated as there was no participant who did not speak Portuguese, information can be found in lines 106 to 107.

DATA COLLECTION

- How was physical activity assessed? Were people asked whether they were directly sedentary or physically active? Or did the researchers use standardized physical activity questionnaires adapted to the local context? If yes, please indicate the questionnaire used and how these data were converted into physical activity categories. On the other hand, if the authors assessed physical activity using accelerometers, this should also be clearly indicated: We assessed it using a standardized and validated questionnaire called the International Physical Activity Questionnary (IPAQ). More information is included in the text on lines 165 through 170.

- How was alcohol consumption estimated? If this has been assessed using questionnaires, this should be indicated - and also the time period covered by this questionnaire. In general, provide more details in the text or supplementary materials about the questionnaires and categories used: Alcohol consumption was also through a validated questionnaire called Alcohol Use Disorder Identification Test (AUDIT), more details are in the text of the document, more information is in the text in lines 170 through 172.

There are several limitations when evaluating this information through questionnaires; therefore, this should be discussed in the limitations section: Suggestion accepted, described in the text in lines 407 through 410.

P.10. What do the authors mean by physical examination? Please clarify: Expression taken from text. Text reworked.

P.10. Is this waist circumference? Indicate in this section the methodology followed to perform these measurements, especially for neck and waist circumference: Information clarified in the text, in lines 137 to 144.

P.10. How did the researchers ensure that participants did not drink sugary drinks or even a snack for eight hours prior to data collection? This seems complicated in a context like this and, given the extremely high prevalence of pre-diabetes reported, it may be the case that several participants have eaten/dranked something in the previous 8 hours. This is a critical question, so I would like to know if the researchers confirmed it (asked?). To participants if they ate/drank in the previous 8 hours at least. Otherwise, this should definitely be addressed in the limitations section: Text reworked in order to clarify the question, in lines 118 to 123.

P.10 and throughout the manuscript, I suggest avoiding referring to the participants as "indigenous peoples" in all cases. Authors may use "participants" or "individuals" in some parts of the article: Suggestion accepted.

P.10. Authors indicate that “changes in capillary blood glucose and/or any of the tests performed were referred to…” Clarify here for the case of other biomarkers that were considered “altered”. Did the researchers also refer people classified as 'obese' to the health post? Please be more specific. Also, indicate the specific cutoff points and references used to classify 'changed' versus normal somewhere before this paragraph (or this paragraph may go after the authors indicate the cutoff points used): Fitted text, available in lines 178 through 183. Thank you for your collaboration.

P.11. "device calibrated automatically." What exactly is this device? Is it a portable blood analyzer like Cardiocheck? Please indicate details about the device and strips used for measuring, if applicable: Complete information has been entered in the text, in lines 154 through 164.

P.11. Do the indicated BMI categories not include individuals with a BMI <18.5? I assume none of the subjects have a BMI <18.5, or were they excluded from the analysis? In any case, this should be indicated in the text: Corrected information in the text in lines 148 and 153 and in tables 1 and 2. BMI <18.5 was considered.

P.11. Clarify what casual blood pressure measurement means in this assessment. Did the authors take only one measurement or more and average them? Also, indicate which "automatically calibrated device" was used to perform these measurements.: Expression “casual” was removed from the text as it was causing misinterpretation. And a translation error occurred. It is not automatically calibrated, it is automatic and calibrated device, complete information in the text in lines 154 to 155.

P.11. What do the authors mean by "The preparation of the indigenous"? Clarify or delete if unnecessary: text reworded, expression has been removed from text.

STATISTICAL ANALYSIS

This topic has been rewritten to better answer editors' questions, on lines 185 through 196.

Indicate in this section which variables were explicitly considered as independent (predictors) and dependent (outcome) and covariates, if any.

P.12. It is not clear what "variables were compared using hypothesis tests" means. Consider indicating the specific hypothesis being tested and the statistical tests used.

P.12. Some sentences in this paragraph are grammatically awkward or incorrect. Consider checking English grammar or syntax: suggestion accepted.

P.12. The authors indicate that “the less relevant collinearities and variables were eliminated”. Which variables were eliminated? How do the authors define "less relevant" variables?

RESULTS

See Tables 1 and 2 when describing the results.

The entire results thread was rewritten to better answer the reviewers' questions in lines 205 through 271.

P.13. How do the authors know about the association between education level and salaried work in this population? Is there any data / study in these populations - or in others in the region - that can refer to where these variables are correlated? There is no clear link between cause and effect to indicate "consequence" here.

P.13. Do the authors mean understandable as expected? Please consider a different word choice: reworded text.

P.13. Indicate what the socioeconomic classification (D, E) means to international readers: reworded text.

P.13 "In relation to body fat, the increase was high (28.3%) and very high (33.3%)." Please clarify what they mean by "the raise was high": reworded text.

P.14 Across the table, be consistent in using %. Some categories have the % symbols and some don't. No matter the choice of authors, but be consistent: Done

p.16. The authors state that “(…) more likely to become pre-diabetic”. This is a cross-sectional study; therefore, the authors cannot actually test causality and therefore cannot claim that X individuals were more likely to become Y. Consider formulating these as associations or as direct observations: Reworded text.

P.17-18. Table 2. The "1" in the entire table for reference seems unnecessary. Authors can indicate the reference in the variable name or somewhere in the methods: Fixed table.

P.18. Authors state that “… only the variables with statistical significance presented in Table 2 were selected”. Do they mean they only include variables that were significantly different between diabetes categories or other categories? Please clarify in the text: Text reworded.

P.19. Rephrase the following sentence, as it is not clear what the authors mean: "... where there was a predominance of women: Xavante (50.8%) (22), Mura (57.8%) (23), Guarani, Kaiowá and Terena (55.8%). " : Text reformulated.

DISCUSSION:

This session was rewritten to better respond to the reviewers' suggestions in lines 281 to 400.

P.20. The authors claim that income explains the consumption of ultra-processed/industrialized foods. Some evidence must be provided to make this claim. If they did not collect dietary data, at least they could refer to other studies in the region where such an association (between ultra-processed foods and diabetes) was found and discuss how this might explain some of the results found.

P.21. Do the authors consider the low frequencies of physical activity categorized as active / very active? Please clarify.

P.22. The authors mention the prevalence of diabetes observed in other studies among indigenous populations in the Americas, but do not provide the references for these studies. This happens in many other parts of the text and, on several other occasions, references are not included in the first mention of the discussed study. Check that references are properly included throughout the text.

P.22. In the sentence “The causes that justify these frequencies are not fully clarified”. Do the authors refer to the causes that explain the frequencies? Consider a different word choice.

P.22. References to the assertion that a number of studies indicate a high prevalence of obesity should be right after this sentence, not at the end of the paragraph.

P.22. The authors claim that excessive consumption of salt and sugar are risk factors for cardiovascular disease, but they do not cite any literature to support this claim.

LIMITATIONS:

Text rewritten to meet reviewers' recommendations in lines 402 through 410.

P.23. I fully agree with the authors that it may be that the default reference values for CVD or diabetes risk categories are not the actual risk values for the population studied; however, if the authors consider this a limitation of their study, they should expand this discussion and cite other literature that has addressed this issue. There is a lot of literature on different cutoff points for Asian populations versus the standard white European population commonly used in surveys.

P.23. The authors indicate that the results may be less accurate than measurements with blood samples. Do they mean "venous" blood samples? Please clarify: Expression taken from text.

P.23. The authors emphasized stress as a likely contributing factor to the high prevalence of DM. If they find this variable to be relevant in the result, they should address this issue in more detail in the discussion, citing the appropriate literature on the subject: Expression taken from text.

---

## [Editor Report · Decision Letter 2]

23 Jul 2021

Glycemic profile and associated factors in indigenous Munduruku , Amazonas.

PONE-D-21-00872R2

Dear Dr. Hanna Lorena Moraes Gomes,

We’re pleased to inform you that your manuscript has been judged scientifically suitable for publication and will be formally accepted for publication once it meets all outstanding technical requirements.

Kind regards,

Fernando Guerrero-Romero, MD, PhD

Academic Editor

PLOS ONE

Additional Editor Comments (optional):

No additional comments
---

## [Editor Report · Acceptance letter]

26 Aug 2021

PONE-D-21-00872R2 

Glycemic profile and associated factors in indigenous Munduruku, Amazonas. 

Dear Dr. Gomes:

I'm pleased to inform you that your manuscript has been deemed suitable for publication in PLOS ONE. Congratulations! Your manuscript is now with our production department. 

Kind regards, 

on behalf of

Dr Fernando Guerrero-Romero 

Academic Editor

PLOS ONE